# A Precision Strategy to Cure Renal Cell Carcinoma by Targeting Transglutaminase 2

**DOI:** 10.3390/ijms21072493

**Published:** 2020-04-03

**Authors:** Soo-Youl Kim, Jeffrey W. Keillor

**Affiliations:** 1Division of Cancer Biology, National Cancer Center, Goyang 10408, Korea; 2Department of Chemistry and Biomolecular Sciences, University of Ottawa, Ottawa, ON K1N 6N5, Canada; jkeillor@uottawa.ca

**Keywords:** clear cell renal cell carcinoma (ccRCC), transglutaminase 2 (TGase 2), p53, therapeutic target

## Abstract

In a recent report, no significance of transglutaminase 2 (TGase 2) was noted in the analyses of expression differences between normal and clear cell renal cell carcinoma (ccRCC), although we found that knock down of TGase 2 induced significant p53-mediated cell death in ccRCC. Generally, to find effective therapeutic targets, we need to identify targets that belong specifically to a cancer phenotype that can be differentiated from a normal phenotype. Here, we offer precise reasons why TGase 2 may be the first therapeutic target for ccRCC, according to several lines of evidence. TGase 2 is negatively regulated by von Hippel-Lindau tumor suppressor protein (pVHL) and positively regulated by hypoxia-inducible factor 1-α (HIF-1α) in renal cell carcinoma (RCC). Therefore, most of ccRCC presents high level expression of TGase 2 because over 90% of ccRCC showed *VHL* inactivity through mutation and methylation. Cell death, angiogenesis and drug resistance were specifically regulated by TGase 2 through p53 depletion in ccRCC because over 90% of ccRCC express wild type p53, which is a cell death inducer as well as a HIF-1α suppressor. Although there have been no detailed studies of the physiological role of TGase 2 in multi-omics analyses of ccRCC, a life-long study of the physiological roles of TGase 2 led to the discovery of the first target as well as the first therapeutic treatment for ccRCC in the clinical field.

## 1. Introduction

Renal cell carcinoma (RCC) is known to comprise a number of different types of cancers in the kidney, including clear cell renal cell carcinoma (ccRCC), papillary RCC, and chromophobe RCC. Among them, ccRCC represents 75% of all RCC cases [1]. Much of what is known about the genetic basis of this disease was learned from the study of inherited forms of RCC [1]. To date, germline mutations of genes have been associated with an increased risk of kidney cancer and have provided the foundation for mapping the genetic pathways altered in RCC [2]. Mutation of these genes such as von Hippel-Lindau (*VHL*) tumor suppressor gene can activate the hypoxia-inducible factor (HIF) pathway [3] and upregulate the phosphoinositide 3-kinase (PI3K)/ mammalian target of rapamycin (mTOR) pathway [4]. Inactivation of pVHL through protein modification also induces signaling of HIF-1α [5] and NF-κB [6]. Those activated pathways have been targeted for treatment, although there are no RCC-specific drugs with a specific therapeutic target. The first line of RCC treatment was selected by targeting universal cancer characteristics, such as the fast growth of cancer, based on growth signaling including mTOR, vascular endothelial growth factor (VEGF), and platelet-derived growth factor (PDGF) pathways. Temsirolimus [7] and everolimus [8] were selected for mammalian target of rapamycin complex 1 (mTORC1) inhibition. Cabozantinib [9], pazopanib [10], and axitinib [11] were selected for VEGFR inhibition by targeting HIF signaling. Sorafenib [12] and sunitinib were selected for PDGFR inhibition [13]. However, the most important limit for therapeutic outcome in ccRCC is recurrent phenotypes with drug resistance [1]. Due to drug resistance, it is difficult to achieve complete tumor remission [14]. There are two major categories of drug resistance, involving mTORC1 inhibition and receptor tyrosine kinase (RTK) (VEGFRs, PDGF-R) inhibition-induced drug resistance. Rapamycin targeting mTORC1 induces mTORC2, which promotes cancer cell survival through Akt activation [15]. Sunitinib targeting RTK induces autophagy through mTORC1 inhibition, which promotes biosynthetic intermediates for cancer survival [16]. Recently it has been shown that a combination treatment of immune-oncologic drugs, such as nivolumab targeting programmed death receptor-1 and ipilimumab targeting cytotoxic T lymphocytes antigen-4 inhibitors, led to a three-month increase of median progression-free survival compared to sunitinib [17]. Although a combination treatment of immune-oncologic drugs improved survival rate, it did not achieve complete tumor remission.

Recently, the clinical proteomic tumor analysis consortium (CPTAC) study on ccRCC was published, including molecular profiling data on 103 ccRCC tumor cases, as well as 83 samples of normal adjacent tissue [18]. The ccRCC proteomics data set included over 11,000 total proteins and over 40,000 phosphopeptides [18]. In addition to that, other analyses applied to these samples included whole exome and genome sequencings, as well as RNA sequencing [18]. Although this ensemble of data regarding ccRCC may explain why ccRCC shows resistance to therapeutics, it did not identify any specific therapeutic target for ccRCC. Although transglutaminase 2 (TGase 2) expression is reported to be high in ccRCC, and TGase 2 inhibition showed significant potential for a therapeutic approach for ccRCC in a series of past studies [19,20,21,22,23], the CPTAC study did not reveal any significance of TGase 2 in ccRCC [18]. These analytical results did not seem to fit, in the overall effort to find a cure for ccRCC. On the contrary, the discovery of TGase 2 as a therapeutic target for ccRCC may indeed be a breakthrough in the search for a cure of this disease. Analysis of ccRCC mutation from the COSMIC database revealed that less than 4% of total ccRCC patients have a p53 mutation [21]. This suggests that p53 is suppressed, even though 96% of ccRCC patients present wild type functional p53. In this review, TGase 2 is analyzed as a precise therapeutic target for ccRCC, considering how TGase 2 expression is increased, why the increase of TGase 2 is specifically related to ccRCC, and what the role of TGase 2 is that makes it critical to ccRCC survival. A key conclusion of this review article is that TGase 2 is a valid target for ccRCC because TGase 2 is the major regulator of functional p53 in ccRCC, and inhibitors that interfere with binding between TGase 2 and p53 show therapeutic potential.

## 2. Currently Suggested Role of TGase 2 in ccRCC—An Overview

### 2.1. TGase 2 in Normal Tissue

TGase 2 has been known as a cross-linking enzyme for about 50 years. Dr. Folk discovered transglutaminase C (TGase C, later *TGM2* gene renamed to TGase 2) from liver cytosol, at a time when Factor XIII was the only known cross-linking enzyme [24]. Cross-linking activity by TGase 2 was considered as a form of inactivation of substrate proteins [25]. Therefore, an increase of TGase 2 activity in tissues was thought to be responsible for an increase of apoptosis for decades [26] until a TGase 2 knock out mouse was shown to also be able to induce apoptosis [27,28]. In fact, the TGase 2 knock out mouse showed normal phenotype unless it was challenged with allergens. The TGase 2 knock out mouse showed significantly reduced inflammation in a unilateral ureteral obstruction model [29]. It also showed a reversal of allergic asthma induced by ovalbumin [30], pulmonary inflammation induced by bleomycin [31], and nephropathy induced by IgA1 [32]. These phenotypic responses suggested that an increase of TGase 2 expression was responsible for defensive mechanisms against foreign material infiltration [33,34]. Drosophila larvae with TGase knock out showed increased mortality after septic injury [35]. TGase 2 is an important effector of early innate immunity, which helps to promote survival against infection. Under this concept, cancer cells may promote survival during anticancer treatment by adopting an advantage from the defensive mechanism role of TGase 2.

### 2.2. TGase 2 in Cancer Tissue

The general role of TGase 2 in cancer may be difficult to clarify because various roles of increased TGase 2 expression have been reported in various cancers, including pancreatic cancer [36], breast cancer [37,38], ovarian cancer [38], and colon cancers [39]. Although the roles of TGase 2 may be diverse in different contexts of cancers, the role of TGase 2 is commonly related to cancer promotion in growth and migration. TGase 2 is induced in cancer cells, which may promote cancer progression by increases of cell growth [6], epithelial–mesenchymal transition (EMT) [40], drug resistance [41,42,43], and cell signaling (reviewed in [44]). There are a numbers of reports that TGase 2 is involved in cancer progression through activity on the cell surface and in the extracellular matrix (ECM), including its role in the regulation of cell-extracellular matrix (ECM) interactions by complex formation with several types of transmembrane receptors such as integrins β1, β3, and fibronectin (reviewed in [45]). TGase 2 plays the role of a co-receptor for integrin-mediated cell binding to fibronectin in cancer cells. Therefore, TGase 2 knock down reduced the cell attachment, migration and invasion in cancer, showing the same effect produced by knock down of fibronectin [46].

TGase 2 regulates cell signaling directly in the signaling by NF-κB [47], HIF-1α [6], and VEGF [48] by regulation of I-κBα, pVHL, and VEGF, respectively. NF-κB activation is commonly observed in cancer cells, where it promotes drug resistance and cell survival [49] and also recruits tumor associated cells by producing inflammatory molecules [50]. In liver cancer cells, TGase 2 induction was observed by TNF-α treatment, which was induced through NF-κB activation [51]. Indeed, the TGase 2 promoter contains binding sites for the NF-κB response element [52]. Later, we were surprised by the opposite regulation, in that TGase 2 was also able to activate NF-κB through I-κBα depletion via polymerization in immortalized glial cells [47] and breast cancer cells [42].

pVHL is a well-known tumor suppressor. *VHL* mutation is often observed in cancer, which results in an increase of the HIF-1α protein level [53] and IGF-1R transcription level [54]. It has been reported that TGase 2 was induced by hypoxia through HIF heterodimer (HIF-1α and HIF-1β) binding at promoter regions of HIF-1α response molecule [55]. However, an increase of TGase 2 expression was also able to induce HIF-1α through pVHL depletion by direct polymerization activity in ovarian cancer and breast cancer cells [6]. As we know that HIF-1α is the master controller of angiogenesis, it follows that TGase 2 mediated HIF-1α stabilization may contribute to angiogenesis promotion.

TGase 2 has also been reported to be involved in EMT in cancer cells through β-catenin activation by c-Src activation through complex formation with fibronectin, TGase 2, and c-Src [56]. This implies that different roles of TGase 2 may arise from differential effects in a context-dependent manner because TGase 2 is expressed in all type of tissues, including cancer, as well as immune cells, extra matrix cells, and endothelial cells. TGase 2 has a specific role in each context (reviewed in [57]). At a biochemical level, the differentiated roles of TGase 2 in a context-dependent manner can be explained by the relative affinity and reactivity of alternative physiological substrates (reviewed in [58]). TGase 2 can recognize a wide variety of substrates having different K_M_ values. When two glutamine donor substrates are present, having low and high K_M_ values, TGase 2 preferentially reacts with the substrate having the lower K_M_ value. For example, in the presence of many proteins in the extracellular matrix, TGase 2 readily reacts with fibronectin, which is bound with high affinity [59]. Once TGase 2 reacts with the glutamine donor substrate protein, the resulting acyl-enzyme intermediate binds an acyl-acceptor substrate, depending on the relative affinity of available amines [60]. For example, when two TGase 2 substrates such as histamine (K_M_: 0.38 mM) [61] and ornithine (K_M_: 1.35 mM) [62] are present in the same concentration, TGase 2 will bind and react preferentially with histamine. In addition to this general control at the molecular kinetic level, TGase 2 reactivity is regulated by many activity effectors, including GTP, ATP, calcium, and conditions including pH and temperature, all of which may affect substrate preference in different cancer types. Taking all these factors into account, it can be seen that TGase 2 may promote different cancers by reaction with different biological targets that are dictated by a precisely designated environment. In ccRCC, TGase 2 specifically interacts with p53 to promote oncogenesis. Although the roles of TGase 2 in cancer are well studied, the subgroup of TGase 2-positive patients in specific cancer types is small [63]. Therefore, from a therapeutic point of view, TGase 2 is commonly considered a chemo-sensitizing target instead of an anti-cancer therapeutic target [42]. However, in ccRCC, TGase 2 is over 90% positive and plays a major role in suppressing wild type p53 [21]. The impact of this correlation is significant, as it indicates that TGase 2 can be a specific target for ccRCC therapy and that inhibitors such as GK921 and streptonigrin, which inhibit binding between TGase 2 and p53, can be an anti-cancer therapeutics [23,64].

### 2.3. TGase 2 in ccRCC

Recently the role of TGase 2 in ccRCC was reviewed, revealing the importance of its role (reviewed in [44]). Clinical data analysis regarding TGase 2 expression in ccRCC patients from The Cancer Genome Atlas showed increased expression of TGase 2 in RNA and protein levels in ccRCC [23,65]. Meta-analysis of ccRCC patients showed that an increased level of TGase 2 was inversely correlated with five-year disease free survival [66]. Furthermore, this study showed that an increase of TGase 2 expression was correlated with an increase of metastatic potential, as well as worse prognosis [65].

It has been shown that TGase 2 knock down induced p53-mediated cell death [19] and that inhibition of TGase 2 by a single treatment of 0.2 mg/kg of streptonigrin showed almost a complete response of therapeutic effect [23] in ccRCC xenograft models. A series of reports showed that in ccRCC cell lines, cell death was induced by TGase 2 knock down using siRNA of TGase 2, while TGase 2 knock down did not lead to any cell death effect in the normal immortalized cell HEK293 [19,20,21,23,67]. This implies that ccRCC specifically employs TGase 2 for survival. TGase 2 acts as a chaperone protein in addition to a cross-linking enzyme [22]. TGase 2 forms a ternary complex wherein the N-terminus of p53 binds to the N-terminus of TGase 2 while the C-terminus of TGase 2 binds the N-terminus of p62 [21].

Additionally, p53 inactivation in cancer is associated with aggressive growth and poor prognosis. Many cancer types have a high level of p53 mutation that abrogates p53’s tumor suppressive role. However, less than 5% of ccRCCs show p53 mutations [21], even though p53 levels are down-regulated [1]. TGase 2 expression depletes p53 by direct binding and transferring to the autophagosome, which results in increased tumor cell survival [19]. TGase 2 targets the DNA-binding domain (residue 102–292 of p53) [21]. Upon deletion of this region, p53 completely loses its capacity to bind to TGase 2 [19]. p53-TGase 2-p62 complexes are specifically transported to the autophagosome and are degraded by the autophagic process [19]. TGase 2-mediated autophagic degradation of p53 is a novel mechanism that does not require TGase 2 catalytic activity, as demonstrated in Figure 1.

TGase 2 inhibitors targeting its catalytic activity did not induce p53 mediated apoptosis in ccRCC. This result concurs with the mechanism of action of streptonigrin, which is bound at the N-terminus of TGase 2, resulting in prevention of p53 binding by competition in the same site and induction of p53-mediated apoptosis by the released p53.

It is known that over 90% of ccRCC shows pVHL inactivation, which leads to an increase of HIF-1α activity [3]. TGase 2 is also induced under hypoxic stress in cancer cells by HIF-1α activation [55]. The TGase 2 promoter region contains a HIF binding region and an NF-κB binding region [52]. It is clear that hypoxia and/or pVHL inactivity are major contributors to TGase 2 induction in ccRCC through HIF-1α activity [55]. Interestingly, pVHL was found to be a target of TGase 2, which can be depleted by cross-linking [6]. This suggests an amplified cycle of HIF-1α signaling, because induced TGase 2 suppresses the pVHL feedback mechanism.

### 2.4. Anti-Cancer Effect of TGase 2 Inhibition in ccRCC

The Keillor group recently reported the TGase 2 inhibitor VA4, optimized from hit compound NC9. This inhibitor blocks the transamidation activity of TGase 2, as well as its GTP binding ability, through an allosteric mechanism [72]. VA4 was reported to be the best lead compound among the sulfonamide and piperazinyl amide derivatives designed to interact with the active site of TGase 2 [72]. We investigated whether these specific irreversible inhibitors for TGase 2 might also block TGase 2—p53 binding and p53 depletion in RCC cells. Our results showed that these inhibitors have no effect on RCC survival (Figure 2). Although these inhibitors were reported to induce a conformational change of TGase 2 that affects GTP binding to TGase 2, the binding of these inhibitors to TGase 2 did not apparently block p53 binding to TGase 2. This result corroborates previous reports regarding p53 binding to TGase 2, in that it is specifically the N-terminus of p53 that interacted with the N-terminus of TGase 2 [21,22,23,64]. This suggests indirectly that streptonigrin induced p53 activation by competing with p53 for binding to TGase 2 [23].

## 3. Precision Strategy for Targeting TGase 2 in RCC

### 3.1. TGase 2 Is a Major VHL-Repressible Gene

Somatic mutation or inactivation via gene modification of both *VHL* tumor suppressor alleles is commonly observed in ccRCC [73]. The gene product pVHL has a ubiquitin ligase activity that targets the HIF-1α subunit by oxygen dependent proteolysis [74]. Therefore, pVHL inactive cells induce HIF-1α stabilization leading to the activation of specific target genes, such as pyruvate kinase or VEGF, involved in the increase of glycolysis or angiogenesis, respectively [75,76]. More than 60 putative direct HIF-1 target genes have been identified. Among them, TGase 2 was observed in the category of amino-acid metabolism [77]. It was also observed in cancer cells that TGase 2 was induced by HIF-1α under hypoxia [55].

There is an interesting report about analysis of *VHL* target genes suppressed by oxygen regulation in ccRCC. In the report, TGase 2 was shown to be induced over 5-fold by hypoxia in ccRCC cells with pVHL expression [68]. This suggests that TGase 2 is one of the top targets of *VHL*. Therefore, the high level of *VHL* inactivation in ccRCC was primarily responsible for the induction of TGase 2 expression (Figure 1).

### 3.2. mir1285 Directly Regulating TGase 2 Suppresses RCC

A group investigating non-coding RNA in cancer investigated microRNA (miRNA) expression signatures and searched for tumor suppressive miRNAs in RCC [69]. They found that the level of *miR-1285* was extensively decreased in RCC. Transfection of *miR-1285* showed a strong inhibitory effect on RCC proliferation [69]. The report showed that 11 genes had target sites for *miR-1285* in their 3′-untranslated regions. Among them, TGase 2 was a target of *miR-1285*. TGase 2 expression was down regulated 4–8 fold by *miR-1285* transfection in RCC (Figure 1) [69]. Furthermore, this report showed that TGase 2 knock down significantly reduced cell proliferation and invasion in RCC cells [69], which corroborates other reports that TGase 2 knock down induced cell death [19,21,23].

### 3.3. TGase 2 Is Induced by mTOR Inhibition

Targeting the mechanistic target of rapamycin complex 1 (mTORC1) raises great concern of drug resistance after treatment, although this treatment may have an anti-proliferative effect on cancer cells. mTORC1 is almost the last signaling kinase in the process of biosynthesis (anabolism), which is absolutely required for cell proliferation [78]. FDA-approved mTORC1 inhibitors, such as Temsirolimus and Everolimus, showed only limited anti-cancer effects and recurrence after treatment because rapamycin and rapalogs are generally cytostatic instead of cytotoxic [78]. There was a report about identification of rapamycin-enhanced transcription changes using a public GEO dataset focusing on rapamycin-treated tuberous sclerosis complex (TSC)-deficient cells. One of the markedly increased transcripts was the *TGM2* gene (TGase 2 protein), which occurred five times in the top 20 probes among 39,000 probed genes [70]. TGase 2 was induced by rapamycin treatment in a time dependent manner (Figure 1) [70]. Although this report did not show the mechanism by which TGase 2 was induced by mTOR inhibition, it showed that combined inhibition of TGase 2 and mTOR reversed mTOR resistance in cancer cell. Autophagy induction by mTOR inhibition or AMPK activation has been reported to have a therapeutic effect in cancer [79]. However, it is also known that mTOR inhibition induces drug resistance through autophagy induction in cancer cells [80]. Interestingly, an increase of TGase 2 is correlated with an increase of autophagy by increase of LC3 levels in various cancer cells [81]. This report suggests that TGase 2 induces drug resistance by potentiating autophagy through LC3 induction via p53 depletion in cancer [81].

## 4. Conclusions

TGase 2 is not critical for normal life because the TGase 2 knock out mouse showed a normal phenotype under normal conditions [28,29,82]. However, the TGase 2 knock out mouse behaved differently from a wild type mouse under stress—including allergic reaction, septic shock, and cancer growth [33]. Loss of TGase 2 expression causes great damage in sepsis but conversely, it helps with evading inflammation in an allergic reaction [29]. In cancer, TGase 2 knock down causes down regulation of cancer progression because the roles of TGase 2 in cancer converge, promoting cancer progression by way of adhesion, proliferation, and EMT (reviewed in [44,83]). TGase 2 knock down or inhibition by single targeting specifically induces an anti-cancer effect in ccRCC through stabilization of p53, because p53 binds directly to TGase 2 and is thus chaperoned to the autophagosome for degradation [19,20,21,23,67,71]. This is only possible in ccRCC because over 90% of p53 in ccRCC is wild type. Thus, collectively these observations represent a breakthrough for the clinical field because they identify TGase 2 as a specific therapeutic target for ccRCC, and the marked potential for binding inhibitors between TGase 2 and p53 such as GK921 [71] and streptonigrin [23] to become the first therapeutics for ccRCC (reviewed in [64]).

## Figures and Tables

**Figure 1 ijms-21-02493-f001:**
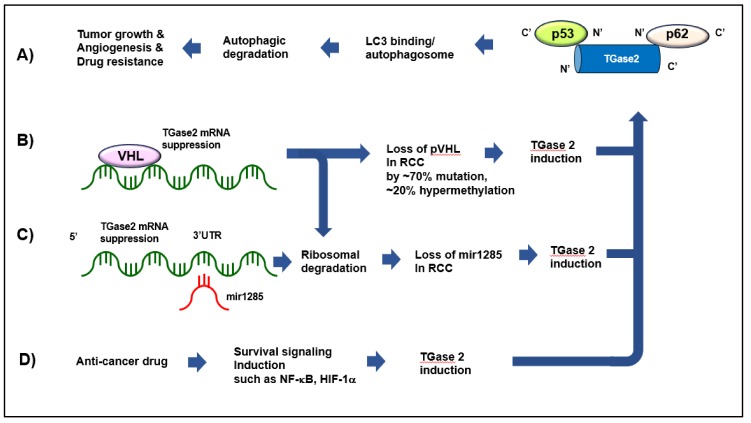
Various pathways inducing TGase 2 expression, as well as the major role of TGase 2 in ccRCC. (**A**) TGase 2 can form a complex by binding p53 and p62 simultaneously, which is then transferred to the autophagosome [21,22]. (**B**) *VHL* inactivation in about 90% of ccRCC induces TGase 2 expression, which can potentiate suppressing p53 signaling [68]. (**C**) Universal decrease of mir1285 level in RCC induces TGase 2 expression, which correlates with RCC proliferation [69]. TGase 2 mRNA expression is suppressed by mir1285 binding at 3′UTR of TGase 2. (**D**) TGase 2 expression is also induced by anti-cancer drug treatment along with drug resistance through NF-κB activation [42,70]. All of these pathways can be reversed by a single treatment of GK921 [71] or streptonigrin [23], which binds to the N-terminus of TGase 2 where p53 would bind, prior to transfer to the autophagosome in ccRCC [64].

**Figure 2 ijms-21-02493-f002:**
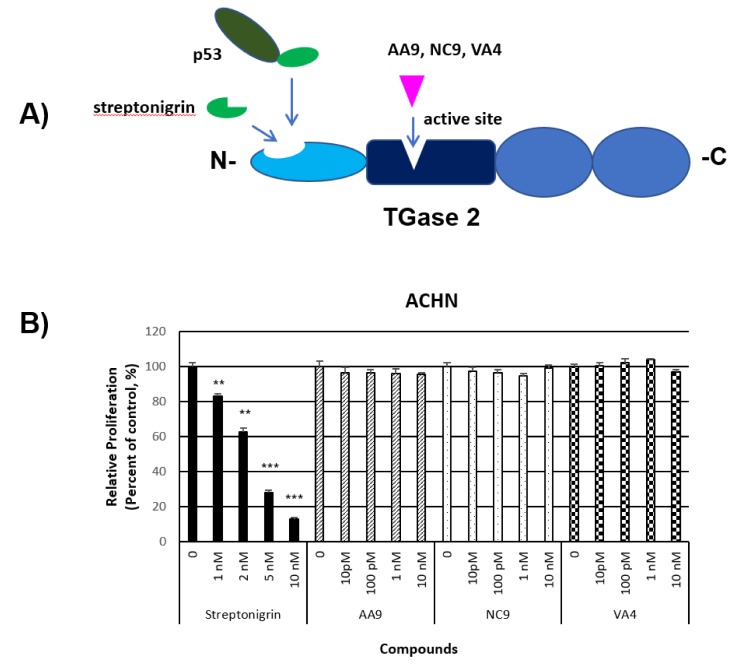
Inhibitor targeting the N-terminus p53 binding region of transglutaminase 2 (TGase 2) shows an anti-cancer effect in clear cell renal cell carcinoma (ccRCC), cell line ACHN. (**A**) Streptonigrin binding to N-terminus of TGase 2 stabilized p53, which induced apoptosis in ccRCC. AA9, NC9, and VA4 binding to active site of TGase 2 caused inactivation of TGase 2. (**B**) Cells were tested for proliferation assay using sulforhodamine B assay as reported [23] for 48 h with the indicated concentrations of streptonigrin, AA9, NC9 and VA4 compounds. Streptonigrin showed an anti-cancer effect in a dose-dependent manner, while TGase 2 active site inhibitors such as AA9, NC9, and VA4 [72] did not show an anti-cancer effect in ccRCC. This suggests that TGase 2 mediated p53 depletion is not dependent on enzyme activity but related to an N-terminal binding chaperone effect to the autophagosome. (mean ± SD, n = 3) ** *p* < 0.01, and *** *p* < 0.001.

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
