# Peer review of "A Precision Strategy to Cure Renal Cell Carcinoma by Targeting Transglutaminase 2"

_ijms, 2020, doi:10.3390/ijms21072493_

Round 1

Reviewer 1 Report

The article entitled “A Precision Strategy to Cure Renal Cell Carcinoma by Targeting Transglutaminase 2” is a review article aiming to summarize their recent findings and convince reader that TGase 2 is a novel target for the treatment of clear cell renal cell carcinoma (ccRCC).

Here are some concerns that need to be addressed before publication:

Major concerns:

  1. It is recommended that the entire manuscript be thoroughly edited for English as it is difficult to read and understand the significance.

Add one extra figure can help to explain things better.

i.e.,.  pVHL, HIF-1a, and miR1285 in regulating TGase 2

  1. As this is a review article, the descriptions should be as comprehensive as possible. Authors are expert in the field of TGase 2 and ccRCC, but not all readers. Sufficient background information should be given before critical new information is introduced. 

Lines 30-45:  The following information should be introduced at the beginning instead of introducing them scattering in the later paragraphs.

  1. What are the major differences between the treatment and p53 status for RCC and ccRCC ?
  2. What are the current treatments for ccRCC?
  3. What are the current target therapies for ccRCC?
  4. What are the major obstacles in treating ccRCC? What contribute to resistance to radiotherapy and chemotherapy in ccRCC ?
  5. What are the current target therapies and their advantage and disadvantages in the treatment?
  6. Beside TGase 2, what other factors are involved in the inactivation of p53 in ccRCC? Why targeting TGase 2 over other factors?
  7. Mutation rate of p53 in ccRCC should be introduced first instead of putting in the conclusion.

  1. Given the complexity and roles of TGase 2 in cancer promotion including EMT, cell adhesion, etc. Authors should clearly present to support why TGase 2 is a breakthrough in the treatment of ccRCC.
  2. Many sentences are not cited with references especially in the conclusion.
  3. In many sentences, RCC is mixed with ccRCC.

Sometimes VHL is also mixed with pVHL

Line 109-121: 

Authors state that TGase 2 interact with specific substrates depending on their KM .

If there is specific substrates, please provide it.

  1. Figure 1 is not clearly presented. There are 1) and 2) subfigure in this figure.  Please use A or B instead to prevent confusion with figure number.

In 2) subfigure, there are no standard deviations for the data.   What is the Y axis represent?  Based on the Figure legend, it is cell proliferation relative to control.   Please revise it.

What is “ACHN” represent on top of Fig. 1  2) figure.

In Fig 1, 1):  Are AA9, NC9, VA4 also tested in the xenograft animal model or just in vitro cell studies.  Please indicate clearly in the figure legends.

  1. Streptonigrin is not a specific inhibitor of TGase 2, How can it be justified as a precision treatment?

Minor:

Line 113:   HUVEC is also an endothelial cell.

Author Response

March 25, 2020

Manuscript ID: ijms-744221
Type of manuscript: Review
Title: A Precision Strategy to Cure Renal Cell Carcinoma by Targeting
Transglutaminase 2
Authors: Soo-Youl Kim *, Jeffrey Keillor

We have modified the manuscript in response to the reviewers’ comments to the best of our ability. Please see the attachment.

Sincerely,

Soo-Youl Kim

Reviewer 2 Report

In this review article, Kim SY and Keillor JW are suggesting the potentiality of Transglutaminase 2 (TGase 2) as a therapeutic target in clear cell – renal cell carcinoma (ccRCC).

Despite great advances in the therapy of ccRCC, current treatments are eventually failing due to the development of cancer resistance. Therefore, it is not trivial to search for potential alternative molecular targets.

Transglutaminase 2 is a well-known cross-linking enzyme which has been described as a mediator of defensive mechanisms against foreign materials (inflammation, allergy) and in the innate immunity response under normal circumstances. TGase 2 has also been described in the context of cancer, promoting cancer progression, EMT and drug resistance.

The Authors report some experimental evidences showing a TGase 2 role in ccRCC cell proliferation. TGase 2 knock-down induced a p53-mediated cell death in xenograft models of ccRCC, and the biochemical regulation of TGase 2 on p53 has been described in a paper published by one of the Authors (Lee, SH et al. Cell Death Dis, 2016).

The inhibition of TGase 2 might have anti-cancer effects in ccRCC. TGase 2 expression is regulated by HIF1α, which is often accumulated in ccRCC upon VHL mutations, being hypoxia a major driver in kidney cancer. Among the possible strategies to inhibit TGase 2, miR-1285 has been described having an inhibitory effect on TGase 2 transcription, able to reduce cell proliferation and invasion of RCC cell lines upon transfection. Finally, TGase 2 expression is induced by mTOR inhibition, a common therapeutic strategy used in ccRCC treatment, often leading to cancer resistance; combined inhibition of mTOR and TGase 2 reversed resistance in RCC cell lines.

This review article has been written by leading experts in the study of the biologic roles of transglutaminases and in the study of their pharmacological characterization (TGase 2 inhibitors). The consequentiality of the paragraphs is quite linear from the main hypothesis (TGase 2 have a role in cancer, specifically in kidney cancer) and might be a potential druggable target in ccRCC, being also implied in the development of resistance (especially after mTOR inhibitors). Anyway, I personally think that sometimes the role of TGase 2 is overstated, or at least not supported by striking experimental evidences. Finally, I personally think that the figures should be implemented, as figure 1 does not help the reader to summarize a key concept, but rather focus on a specific experimental evidence. Moreover, I think an additional figure might be added to summarize the TGase 2/Nf-κB axis and the TGase 2/VHL/HIF axis.

Coming to a more detailed examination of the manuscript I have some concerns:

Lines 37-38 – what do you mean by “there are no RCC-specific drugs having a specific therapeutic target”? This is not true. The drugs used in treatment of RCC are small molecules, in the large majority. Some of them have a broad target kinome, Sorafenib, while others are quite specific it their action, Axitinib. Not to forget an entire class of molecules, monoclonal antibodies, which are also used.

Line 42 – “carbozantinib” is cabozantinib. This is quite a peculiar drug since it has an inhibitory effect on both VEGFR receptor an c-MET. In this collection of available treatments, anyway, a major group is missing and should be also taken into account: the anti-PD1 and anti-PD-L1 monoclonal antibodies that introduced a real breakthrough also in the context of ccRCC treatment.

Lines 55-56 – Since the results from the CPTAC database do not reveal any significance in TGase 2 in ccRCC (protein expression levels? Compared to normal adjacent tissue?), I would try to rephrase the sentence stating that TGase 2 might be a breakthrough in ccRCC treatment, but I would otherwise comment on this discrepancy already at this point, or at least discuss it in paragraph 2.3.

Lines 78-79 – The assumption that the overexpression of TGase 2 in a mouse model might increase tumor development risk with the accumulation of mutations by aging is an hypothesis that is overstating the role of the enzyme, not supported by experimental evidences.

Lines 84-86 – “cancer promotion in cancer cells”, I understand the meaning, but the sentence should be rephrased as it does not make sense from a biological point of view.

Lines 92-93 – if a knock-down of TGase 2, described as a co-receptor of integrin-mediated cell binding to fibronectin, reduced cell adhesion, I would expect an increase in migration or invasion. Does anoikis have a role? Can you include a comment on this point?

Lines 103-104 – this sentence is not very clear: which protein is binding the promoter region of TGase 2 after the hypoxic stimulus?

Lines 112 – if TGase 2 is expressed in all type of tissues, how would you comment on the precision of TGase 2 inhibition as a cancer treatment? I assume context-dependency might be an explanation, but the following paragraph on the pharmacokinetics of the enzyme does not explain it and does not show any reference.

Line 130 – I would add a few words on the mechanism of action and the structure of streptonigrin, is it specifically interacting with TGase 2 or does it have multiple effects on other targets?

Line 137 – I think the Authors wanted to write p62 instead of p63.

Line 206 – mTOR inhibitors have not only entered into clinical trials, a few of them (Temsirolimus and Everolimus) are approved therapies for RCC treatment by the FDA and the EMA.

Lines 216-217 – since it is a controversial statement on the role of autophagy in this context, I would add some more comments to discuss this issue.

Author Response

March 25, 2020

Manuscript ID: ijms-744221
Type of manuscript: Review
Title: A Precision Strategy to Cure Renal Cell Carcinoma by Targeting
Transglutaminase 2
Authors: Soo-Youl Kim *, Jeffrey Keillor

Dear Editors,

We have modified the manuscript in response to the reviewers’ comments to the best of our ability. Please see the attachment.

Sincerely,

Soo-Youl Kim

Round 2

Reviewer 1 Report

The manuscript is much improved and only minor spelling check is recommended.

Author Response

no comment.

Reviewer 2 Report

In this revised review article, Kim SY and Keillor JW are suggesting the potentiality of Transglutaminase 2 (TGase 2) as a therapeutic target in clear cell – renal cell carcinoma (ccRCC).

Overall, the manuscript has been largely improved. I personally think that the paragraph suggesting how context dependency of TGase 2 activity can be explained by pharmacokinetics has given more rationale to the possible role of TGase 2 as an anti-cancer therapeutic in ccRCC.

The introduction of Figure 2 is also very valuable and helps the reader to summarize the mechanism by which TGase 2 is induced in ccRCC.

However, I still have a few minor concerns:

  • Lines 37-38 – thank you for the explanation in the rebuttal letter. To make it clearer to the audience, I would add it in the manuscript in between brackets (the example of Her2 for breast cancer or BCR-Abl for CML makes the concept clearer).
  • Line 16 – “the first” can be changed into “a valuable/an extremely valuable” since I think it is best to avoid overstatements. Or do you mean “first line”?
  • Line 18 – missing “)” after HIF-1α.
  • Line 19 – “present” rather than “presents”.
  • Line 74 – “the major regulator of p53”, I do not agree with this since it questionable, many other proteins have a major role in p53 regulation (MDM2 as an example). I would just change the adjective to be more cautious.
  • Line 101 – “a numbers of” better “a number of” or “many”.
  • Line 108 – “TGase 2 regulates cell signaling directly in the signaling by NF-κB, HIF-1α and VEGF 
by regulation of I-κBα, pVHL and VEGF respectively” this sentence is not clear.
  • Line 123-150 – I personally think this is a very good paragraph but I would just leave out the sentence from lines 142-143 “In ccRCC, TGase 2 specifically interacts with p53 to promote oncogenesis”.
  • Line 267 – “first” still, not clear. Do you mean “first line”. If I understood correctly, TGase 2 might be first line therapeutic target better than the others commonly in use which are not specific for RCC?
  • Figure 2 – I personally would move the A) section to the D) section. It is more linear. The reader might be puzzled by the reading right to left the first point of the figure, but might accept it better as a last point.

Author Response

All minor concerns are up to author's style.

If I have to make corrections on spelling or grammar, I will do it during galley proof.

Please respect author's contributions.

Soo